# Nevanlinna.jl: A Julia implementation of Nevanlinna analytic continuation

Kosuke Nogaki[1*], Jiani Fei[2,3], Emanuel Gull[3] and Hiroshi Shinaoka[4,5],

**1** Department of Physics, Kyoto University, Kyoto 606-8502, Japan
**2** Department of Physics, Stanford University, Stanford, CA 94305, USA
**3** Department of Physics, University of Michigan, Ann Arbor, MI 48104, USA
**4** Department of Physics, Saitama University, Saitama 338-8570, Japan
**5** JST, PRESTO, 4-1-8 Honcho, Kawaguchi, Saitama 332-0012, Japan
* nogaki.kosuke.83v@st.kyoto-u.ac.jp

September 20, 2023

## Abstract

We introduce a Julia implementation of the recently proposed Nevanlinna analytic continuation method. The method is based on Nevanlinna interpolants and inherently preserves the causality of a response function due to its construction. For theoretical calculations without statistical noise, this continuation method is a powerful tool to extract real-frequency information from numerical input data on the Matsubara axis. This method has been applied to first-principles calculations of correlated materials. This paper presents its efficient and full-featured open-source implementation of the method including the Hamburger moment problem and smoothing.

# 1   Introduction

In finite-temperature quantum field theories ranging from condensed matter to high-energy physics, many sophisticated numerical techniques have been developed. For instance, per­turbative theories [1–7] are a powerful tool for studying impurity effects [8], Fermi liquids [9, 10], and symmetry breaking phenomena such as charge-, spin-density waves [11–13], or superconductivity [14, 15]. For investigating Mott transitions and renormalization effects of quasiparticles near the Fermi energy, or Kondo effects, we may employ the non-perturbative dynamical mean-field theory [16] with discrete- [17] or continuous-time [18–21] quantum Monte Carlo impurity solvers. In the field of high-energy physics, lattice quantum chromo­dynamics algorithms are used for *ab initio* investigations of the masses of hadrons, the quark confinement, or of chiral symmetry breaking [22–24].

These theories are formulated in "imaginary time", where finite-temperature statistical mechanics computations are tractable. The result of the computation is the numerical data of the Matsubara Green's function $\mathcal{G}(i\omega_n)$ defined on the imaginary axis of the complex frequency plane. The spectral function $\rho(\omega) = -(1/\pi)\mathrm{Im}G^{\mathrm{R}}(\omega)$ contains information about the single-particle excitation which, in electronic systems, are related to measurements in photoemission spectroscopy. An analytic continuation step relating the Matsubara Green's function $\mathcal{G}(i\omega_n)$ to the retarded Green's function $G^{\mathrm{R}}(\omega)$ is therefore needed as a post-processing step. This need for numerical analytic continuation exists not only for fermionic systems but also for bosonic systems [25, 26] including He [27, 28], supersolids [29], and warm dense matter [30]. Thus, a highly precise and efficient numerical analytic continuation method is desired for quantitative studies of quantum many-body systems.

Regardless of its practical importance, the numerical analytic continuation of the Green's function is an ill-conditioned problem whose direct solution is intractable. To address this is­sue, many approximate methods have been developed. Examples include continued fraction Padé approximation methods [31], the maximum entropy method [32, 33], the stochastic an­alytic continuation [34–38], machine learning approaches [39], genetic algorithms [28], the sparse modeling method [40, 41], the Prony method [42], and a pole fitting approach [43]. Most of these methods are based on a regularized fit and fail to restore sharp structures in the large-$\omega$ region even for numerically exact input data. The Padé approximation, which is an interpolation method, does not ensure causality and often results in negative values of the spectral function and a violation of the sum rule, particularly at high frequencies.

The Nevanlinna analytic continuation method [44], an interpolation method, inherently respects the mathematical structure of causal response functions, thereby providing a mathe­matically rigorous numerical analytic continuation that ensures causality. The formalism has been extended to matrix-valued Green's functions [44, 45].

While the Nevanlinna analytic continuation method has an elegant mathematical founda­tion, the numerical solution of the Nevanlinna continuation equations requires special care.

The continued fraction expressions used in the method are sensitive to numerical precision, which means that the interpolation must be performed using at least quadruple floating-point arithmetic, even if input data is only known to double precision. In addition, selecting a subset of the input data such that it respects the so-called Pick condition, which guarantees causality [44], is essential to avoid overfitting. For a solvable non-degenerate problem, Nevanlinna theory guarantees the existence of an infinite number of valid analytical continuations. In practical applications, a single "best" one of these needs to be chosen, typically by imposing an additional smoothness constraint.

The sample `C++` code published by the authors of [44] as a supplement to the original paper serves to illustrate Nevanlinna continuation but does not implement this smoothing step or a selection algorithm for choosing a subset of causal data. In this paper, we describe a full-featured implementation of the Nevanlinna analytic continuation method in the Julia language. Our implementation incorporates interpolation executed in arbitrary-precision arithmetic, which ensures a stable interpolation. We execute the smoothing based on numerical optimization, utilizing the automatic differentiation of the cost function, which is faster and more accurate than the numerical finite difference method. The code is straightforward to install and comes with Jupyter Notebooks illustrating typical use cases. The implementation in the Julia language makes the code easily customizable for future extensions, e.g., to matrix-valued Green's functions [45]. We expect that providing the user community with a ready-to-use and simple package that implements these additional steps will accelerate the adoption of the Nevanlinna method in finite temperature Green's function calculations.

## 2 Theory

In the Nevanlinna analytic continuation, the analytic properties of Green's function play an essential role. We, therefore, describe the analytic structure of both Matsubara Green's function and the retarded Green's function, focusing on the Lehmann representation in Sec. 2.1. In Sec. 2.2, the definition of Nevanlinna functions is given. Green's functions as Nevanlinna functions, the Pick criterion, and the Schur interpolation algorithm are summarized, and the Hardy optimization procedure is explained with some technical remarks. The fundamental principles of the Hamburger moment problem are presented in Sec. 2.3. For the purpose of constructing a solution, the Hankel matrix and two distinct types of polynomials are introduced. The theory outlined here follows Refs. [44–46]. Additional technical and theoretical details explained in this paper may be useful for users of the code.

### 2.1 Analytic continuation from Matsubara frequency to real frequency

In this paper, we focus on correlation functions between the fermionic annihilation operator, $\hat{c}$, and the creation operator, $\hat{c}^\dagger$, which we call Green's function. The Matsubara Green's function and retarded Green's function are defined as follows:

$$\mathcal{G}(\tau) = -\langle T_\tau \hat{c}(\tau)\hat{c}^\dagger(0)\rangle, \tag{1}$$

$$G^{\mathrm{R}}(t) = -i\theta(t)\langle\{\hat{c}(t),\hat{c}^\dagger(0)\}\rangle. \tag{2}$$

in the imaginary-time domain and in the real-time domain, respectively. Their Fourier-transformed functions are given by

$$\mathcal{G}(i\omega_n) = \int_0^\beta d\tau\, e^{i\omega_n \tau} \mathcal{G}(\tau), \tag{3}$$

$$G^{\mathrm{R}}(\omega) = \lim_{\eta \to +0} \int_{-\infty}^\infty dt\, e^{i\omega t - \eta t} G^{\mathrm{R}}(t) \quad (\eta > 0), \tag{4}$$

where $\langle \cdots \rangle = \mathrm{tr}\{e^{-\beta(\hat{H}-\mu\hat{N})}\cdots\}/\Xi$, $\hat{c}(\tau) = e^{(\hat{H}-\mu\hat{N})\tau}\hat{c}e^{-(\hat{H}-\mu\hat{N})\tau}$, and $\hat{c}(t) = e^{i(\hat{H}-\mu\hat{N})t}\hat{c}e^{-i(\hat{H}-\mu\hat{N})t}$ with Hamiltonian $\hat{H}$, particle number operator $\hat{N}$, the inverse temperature $\beta = 1/T$, and the chemical potential $\mu$. We here set the Boltzmann constant $k_{\mathrm{B}}$ equal to 1. Here, $\Xi = \mathrm{tr}\{e^{-\beta(\hat{H}-\mu\hat{N})}\}$ is the partition function and $i\omega_n = i(2n+1)\pi T$ are fermionic Matsubara frequencies [47]. These two Green's functions are related by the Lehmann representation [48–51],

$$G(z) = \int_{-\infty}^\infty d\omega\, \frac{\rho(\omega)}{z - \omega}. \tag{5}$$

Namely, the Matsubara Green's function $\mathcal{G}(i\omega_n)$ is given by the limit $z \to i\omega_n$, and the retarded Green's function $G^{\mathrm{R}}(\omega)$ is given by the limit $z \to \omega + i\eta$ ($\eta \to +0$). Here, the spectral function $\rho(\omega)$ is

$$\rho(\omega) = \frac{1}{\Xi} \sum_{n,m} e^{-\beta(E_n - \mu N_n)} (1 + e^{-\beta\omega}) |\langle n|\hat{c}|m\rangle|^2 \delta(\omega - E_m + E_n + \mu), \tag{6}$$

where $E_n$ and $N_n$ are energy and particle number of eigen state $|n\rangle$. From this definition, we see that the spectral function $\rho(\omega)$ is always non-negative ($\rho(\omega) \geq 0$), and it satisfies the sum rule:

$$\int \rho(\omega)\,d\omega = \frac{1}{\Xi} \sum_{n,m} \left(e^{-\beta(E_n - \mu N_n)} + e^{-\beta(E_m - \mu N_m)}\right) \langle n|\hat{c}|m\rangle \langle m|\hat{c}^\dagger|n\rangle \tag{7}$$

$$= \left\langle \{\hat{c}, \hat{c}^\dagger\} \right\rangle = 1. \tag{8}$$

Using the following formula

$$\lim_{\eta \to +0} \int \frac{f(x)}{x + i\eta}\,dx = \mathrm{P} \int \frac{f(x)}{x}\,dx - i\pi f(0), \tag{9}$$

the spectral function can be evaluated from retarded Green's function,

$$\rho(\omega) = \lim_{\eta \to +0} -\frac{1}{\pi} \mathrm{Im}\, G^{\mathrm{R}}(\omega + i\eta). \tag{10}$$

The central objective in this paper is to estimate $G^{\mathrm{R}}(\omega + \eta)$ and $\rho(\omega)$ from the data of $\mathcal{G}(i\omega_n)$, namely, *numerical analytic continuation* between $G^{\mathrm{R}}(\omega + \eta)$ and $\mathcal{G}(i\omega_n)$.

## 2.2 Nevanlinna analytic continuation procedure

### 2.2.1 Definition and notations

First, let us summarize the notations used in this paper. The upper half-plane $\mathcal{C}^+$ and the open unit disk $\mathcal{D}$ are

$$\mathcal{C}^+ = \{z \in \mathbb{C} \mid \mathrm{Im}\, z > 0\}, \tag{11}$$

$$\mathcal{D} = \{w \in \mathbb{C} \mid |w| < 1\}. \tag{12}$$

Their closures are denoted by $\overline{\mathcal{C}^+}$ and $\overline{\mathcal{D}}$, respectively. Nevanlinna functions are holomorphic functions from $\mathcal{C}^+$ to $\overline{\mathcal{C}^+}$, and Schur functions are holomorphic functions from $\mathcal{D}$ to $\overline{\mathcal{D}}$. We denote the set of Nevanlinna functions and that of Schur functions as $\mathcal{N}$ and $\mathcal{S}$, respectively. Note that an one-to-one correspondence exists between $z \in \mathcal{C}^+$ and $w \in \mathcal{D}$ by Möbius transformation $h_\xi$ and the inverse $h_\xi^{-1}$ for $\xi \in \mathcal{C}^+$:

$$w = h_\xi(z) = \frac{z - \xi}{z - \xi^*}, \tag{13}$$

$$z = h_\xi^{-1}(w) = \frac{w\xi^* - \xi}{w - 1}. \tag{14}$$

Another Möbius transformation maps $w \in \mathcal{D}$ to $w' \in \mathcal{D}$ for $\zeta \in \mathcal{D}$:

$$w' = g_\zeta(w) = \frac{w + \zeta}{1 + \zeta^* w}, \tag{15}$$

$$w = g_\zeta^{-1}(w') = \frac{w' - \zeta}{1 - \zeta^* w'}. \tag{16}$$

### 2.2.2 Green's functions as Nevanlinna functions

As discovered in Refs. [44, 45], the negative of the fermionic Green's function is a Nevanlinna function. Indeed, from Eq. (5),

$$
\begin{aligned}
G(x + iy) &= \int_{-\infty}^{\infty} d\omega \, \frac{\rho(\omega)}{x + iy - \omega} \\
&= \int_{-\infty}^{\infty} d\omega \, \frac{\rho(\omega)(x - \omega - iy)}{(x - \omega)^2 + y^2}.
\end{aligned}
\tag{17}
$$

Given that $\rho(\omega) \geq 0$,

$$-\mathrm{Im}\, G(x + iy) = \int_{-\infty}^{\infty} d\omega \, \frac{\rho(\omega) y}{(x - \omega)^2 + y^2} \geq 0, \tag{18}$$

where proves $-G(z) \in \mathcal{N}$. In numerical analysis, we can determine the values of Green's function at a finite number of Matsubara frequencies, represented as $-G(Y_\alpha) = C_\alpha \, (\alpha = 1, 2, \ldots, M)$. The problem is to find Nevanlinna functions $f \in \mathcal{N}$ which satisfy $f(Y_\alpha) = C_\alpha$. This problem can be modified into another tractable problem by transforming the range of Nevanlinna function by Möbius transformation. Therefore, our problem is to find a composite function $\theta = h_i \circ f : C^+ \to \overline{\mathcal{D}}$ which satisfy $h_i \circ f(Y_\alpha) = h_i(C_\alpha) = \lambda_\alpha$. We call these modified Nevanlinna functions contractive functions. As discussed below, interpolation problems of contractive functions can be solved efficiently by the Schur algorithm [52, 53].

### 2.2.3 Pick criterion

There is a necessary and sufficient condition for the existence of Nevanlinna interpolants, namely, the generalized Pick criterion [44]. It is formulated in terms of the Pick matrix [54],

$$\left[ \frac{1 - \lambda_\alpha \lambda_\beta^*}{1 - h_i(Y_\alpha) h_i(Y_\beta)^*} \right]_{\alpha, \beta} \qquad \alpha, \beta = 1, 2, \ldots, M. \tag{19}$$

and states that if the Pick matrix is positive definite, an infinite number of solutions to the interpolation problem exists. If it is positive semidefinite but not positive definite, there is a

unique solution. If the Pick matrix contains negative eigenvalues in addition to the positive ones, no solution to the interpolation problem exists [54, 55]. In many numerical calculations, this condition is satisfied when considering a subset of the values to be interpolated, but it fails when all values are taken into account. In particular, if data points are added from low to high frequencies, high Matsubara frequency values tend to break this condition. Our implementation determines the *optimal* number of low Matsubara frequencies, $N_{\mathrm{opt}}$, for the analytic continuation in an automated fashion. The process involves setting an initial value of $N_{\mathrm{cut}}$ to 1 and constructing a Pick matrix from input data at the lowest $N_{\mathrm{cut}}$ Matsubara frequencies ($\alpha = 1, \cdots, N_{\mathrm{cut}}$), which is then factorized using Cholesky Factorization. If the factorization is successful, $N_{\mathrm{cut}}$ is incremented by one and the procedure is repeated until a factorization failure occurs [1]. The optimal cutoff, $N_{\mathrm{opt}}$, is then determined as the maximum value of $N_{\mathrm{cut}}$ for which factorization is successful. In the subsequent analytic continuation, we utilize only the data up to $N_{\mathrm{opt}}$. We refer to this procedure as "Pick Selection".

### 2.2.4 Schur algorithm

The numerical analytic continuation can be viewed as a problem of constructing an analytic function subject to $M$ point constraint conditions. That is, we aim to construct a contractive function that satisfies

$$\theta(Y_\alpha) = \lambda_\alpha \quad (\alpha = 1, 2, \ldots, M). \tag{20}$$

The Schur Algorithm iteratively interpolates and constructs $\theta(z)$. In the following, we begin by constructing a contractive function with a single constraint. This process will subsequently be generalized to accommodate $M$ constraint conditions.

First, let us consider a Schur function $\varphi \in \mathcal{S}$ with one constraint condition $\varphi(0) = \gamma_1 \in \mathcal{D}$. We construct the function

$$\tilde{\varphi}(w) = \frac{1}{w} \frac{\varphi(w) - \gamma_1}{1 - \gamma_1^* \varphi(w)} \tag{21}$$

$$= \frac{1}{w} g_{\gamma_1}^{-1}(\varphi(w)). \tag{22}$$

From $g_{\gamma_1}^{-1}(\varphi(0)) = 0$ and the Schwartz's lemma, $\tilde{\varphi}(w)$ belongs to $\mathcal{S}$. Conversely for any Schur function $\tilde{\varphi}(w)$,

$$\varphi(w) = \frac{w\tilde{\varphi}(w) + \gamma_1}{1 + \gamma_1^* w\tilde{\varphi}(w)} \tag{23}$$

$$= g_{\gamma_1}(w\tilde{\varphi}(w)) \tag{24}$$

will be regular in $\mathcal{D}$, $|\varphi(w)| < 1$, and $\varphi(0) = \gamma_1$. Therefore, Eq. (23) provides a general form of Schur functions subject to a single constraint condition $\varphi(0) = \gamma_1$, where $\tilde{\varphi}(w)$ is an arbitrary Schur function.

Combining Eq. (23) and Möbius transformation $h_{Y_1}(z)$, a general form of contractive functions $\theta(z) = \varphi \circ h_{Y_1}(z)$ that satisfy $\theta(Y_1) = \gamma_1$ an be given as

$$\theta(z) = \frac{\frac{z - Y_1}{z - Y_1^*} \tilde{\theta}(z) + \gamma_1}{\gamma_1^* \frac{z - Y_1}{z - Y_1^*} \tilde{\theta}(z) + 1}, \tag{25}$$

where $\tilde{\theta}(z)$ is an arbitrary contractive function.

---

[1]Rigorously speaking, the success of Cholesky factorization does not guarantee that the given matrix is positive definite due to rounding error. The numerical rigorous criterion can be found in Ref. [56]

The procedure can be further extended to problems with $M$ constraint conditions:

$$\theta_1(Y_\alpha) = \lambda_\alpha^{(1)}. \qquad (\alpha = 1, 2, \ldots, M) \tag{26}$$

By utilizing Eq. (25), we can recast the $M$ constraint problem for $\theta_1$ as an $(M-1)$ constraint problem for $\theta_2$:

$$\theta_1(z) = \frac{\frac{z-Y_1}{z-Y_1^*}\theta_2(z) + \lambda_1^{(1)}}{(\lambda_1^{(1)})^* \frac{z-Y_1}{z-Y_1^*}\theta_2(z) + 1}, \tag{27}$$

with

$$\theta_2(Y_\alpha) = \frac{Y_\alpha - Y_1^*}{Y_\alpha - Y_1} \frac{\lambda_1^{(1)} - \lambda_\alpha^{(1)}}{(\lambda_1^{(1)})^* \lambda_\alpha^{(1)} - 1} \equiv \lambda_\alpha^{(2)} \quad (\alpha = 2, 3, \cdots, M). \tag{28}$$

In a similar manner, this algorithm can be continued iteratively until $\theta_1$, $\theta_2$, $\cdots$, $\theta_M$, $\theta_{M+1}$ are determined, leaving $\theta_{M+1}$ as an arbitrary contractive function. The continued contractive function, which is parameterized by $\theta_{M+1}$, can be expressed as

$$\theta(z)[\theta_{M+1}(z)] = \frac{a(z)\theta_{M+1}(z) + b(z)}{c(z)\theta_{M+1}(z) + d(z)}, \tag{29}$$

where $a(z)$, $b(z)$, $c(z)$, and $d(z)$ are determined by

$$\begin{pmatrix} a(z) & b(z) \\ c(z) & d(z) \end{pmatrix} = \prod_{\alpha=1}^{M} \begin{pmatrix} \frac{z-Y_\alpha}{z-Y_\alpha^*} & \phi_\alpha \\ \phi_\alpha^* \frac{z-Y_\alpha}{z-Y_\alpha^*} & 1 \end{pmatrix}$$
$$= \begin{pmatrix} \frac{z-Y_1}{z-Y_1^*} & \phi_1 \\ \phi_1^* \frac{z-Y_1}{z-Y_1^*} & 1 \end{pmatrix} \begin{pmatrix} \frac{z-Y_2}{z-Y_2^*} & \phi_2 \\ \phi_2^* \frac{z-Y_2}{z-Y_2^*} & 1 \end{pmatrix} \cdots \begin{pmatrix} \frac{z-Y_M}{z-Y_M^*} & \phi_M \\ \phi_M^* \frac{z-Y_M}{z-Y_M^*} & 1 \end{pmatrix}. \tag{30}$$

Here, $\phi_\alpha$ $(\alpha = 1, 2, \cdots, M)$ is defined by $\phi_\alpha \equiv \theta_\alpha(Y_\alpha) = \lambda_\alpha^{(\alpha)}$. The retarded Green's function $G^R(\omega + i\eta)$ is given by $-h_i^{-1}(\theta(\omega + i\eta))$

To determine $\phi_\alpha$, we prepare the recursive algorithm. First, $\phi_1 = \theta(Y_1)$ and construct

$$\begin{pmatrix} a_2 & b_2 \\ c_2 & d_2 \end{pmatrix} = \begin{pmatrix} \frac{Y_2-Y_1}{Y_2-Y_1^*} & \phi_1 \\ \phi_1^* \frac{Y_2-Y_1}{Y_2-Y_1^*} & 1 \end{pmatrix}, \tag{31}$$

and determine $\phi_2$

$$\phi_2 = \frac{-d_2\theta(Y_2) + b_2}{c_2\theta(Y_2) - a_2}. \tag{32}$$

Generally, the values of $\phi_1, \cdots, \phi_{\beta-1}$ are used to determine $\phi_\beta$ as follows:

$$\begin{pmatrix} a_\beta & b_\beta \\ c_\beta & d_\beta \end{pmatrix} = \prod_{\alpha=1}^{\beta-1} \begin{pmatrix} \frac{Y_\beta-Y_\alpha}{Y_\beta-Y_\alpha^*} & \phi_\alpha \\ \phi_\alpha^* \frac{Y_\beta-Y_\alpha}{Y_\beta-Y_\alpha^*} & 1 \end{pmatrix}, \tag{33}$$

$$\phi_\beta = \frac{-d_\beta\theta(Y_\beta) + b_\beta}{c_\beta\theta(Y_\beta) - a_\beta}. \tag{34}$$

Note that these algorithms require at least quadruple floating-point precision to achieve accurate continued fraction expressions, as numerical instability may arise. This is demonstrated in Section 3.3.

### 2.2.5 Smoothing

There is an infinite number of "valid" continuations consistent with causal input data since any Schur function $\theta_{M+1}$ will yield a valid spectral function. To select the "most physical" of all possible spectral functions, additional constraints for $\theta_{M+1}$ or for the final spectral function can be imposed. As discussed in the following section, artificial oscillations around exact values manifest for $\theta_{M+1}(z) = 0$. To eliminate these oscillations and get the *best* continued result, we adjust $\theta_{M+1}(z)$ in order to get the smoothest possible spectral function [44]. We assume that $\theta_{M+1}(z)$ exists in Hardy space $H^2(\mathcal{C}^+)$ in which a function $F(z)$ satisfies [57]

$$\sup_{y>0} \int_{-\infty}^{\infty} |F(x+iy)|^2 \, dx < \infty. \tag{35}$$

This space is generated by the orthogonal basis $\{f^k(z)\}_0^\infty$ whose basis functions are given by

$$f^k(z) = \frac{1}{\sqrt{\pi}(z+i)} \left( \frac{z-i}{z+i} \right)^k. \tag{36}$$

We expand $\theta_{M+1}(z)$ into the basis with a cutoff parameter $H_{\text{cut}}$,

$$\theta_{M+1}(z) = \sum_{k=0}^{H_{\text{cut}}} a_k f^k(z) + b_k \left[ f^k(z) \right]^*, \tag{37}$$

and minimize the cost function

$$F[\theta_{M+1}] = \left| 1 - \int_{-\infty}^{\infty} \rho(\omega) \, d\omega \right|^2 + \lambda \int_{-\infty}^{\infty} (\rho''(\omega))^2 \, d\omega. \tag{38}$$

Typically, a value of $\lambda = 10^{-4}$ tends to yield stable solutions. In `Nevanlinna.jl`, we use automatic differentiation to optimize coefficients $a_k$, $b_k$. The implementation is based on `Zygote.jl` [58] and `Optim.jl` [59]. The automatic differentiation is extraordinarily efficient and accurate up to machine precision, unlike the numerical finite difference method employed in Ref. [44].

   In practical calculations, a large $H_{\text{cut}}$ can lead to numerical instabilities. As such, our methodology adopts a *step-by-step* approach. Once a solution $(a_k, b_k)$ converges for a given $H_{\text{cut}}$, we initiate the optimization of the cost function for $H_{\text{cut}} + 1$, using the previously converged values $(a_0, \cdots, a_{H_{\text{cut}}}, 0, b_0, \cdots, b_{H_{\text{cut}}}, 0)$ as the initial values. The code commences with an initial cutoff value of $H_{\text{min}}$, and the optimization procedure is repeated by incrementing $H_{\text{cut}}$ until optimization fails. At that point, continued values are computed based on the last converged solution. It is crucial to carefully consider the value assigned to $H_{\text{min}}$, as in certain circumstances, utilizing $H_{\text{min}} = 0$ can fail at the first optimization step. Hence, the optimal value of $H_{\text{min}}$ that leads to convergence should be adopted in such cases.

### 2.3 Hamburger moment problem

The prior knowledge of the moments of the spectral function can be incorporated into the Nevanlinna analytic continuation procedure [46, 60]. The $n$-th moment is defined as

$$h_n \equiv \int d\omega \, \omega^n \rho(\omega). \tag{39}$$

These moments are related to the asymptotic expansion of the Green's function:

$$G(z) = \int_{-\infty}^{\infty} d\omega \frac{\rho(\omega)}{z - \omega} \tag{40}$$

$$= \frac{1}{z} \int_{-\infty}^{\infty} d\omega \frac{\rho(\omega)}{1 - \left(\frac{\omega}{z}\right)} \tag{41}$$

$$= \frac{1}{z} \int_{-\infty}^{\infty} d\omega \sum_{n=0}^{\infty} \left(\frac{\omega}{z}\right)^n \rho(\omega) \tag{42}$$

$$= \frac{h_0}{z} + \frac{h_1}{z^2} + \frac{h_2}{z^3} + \cdots \quad (|z| \to \infty). \tag{43}$$

The correct high-frequency behavior is usually enforced by Matsubara points at large Matsubara frequencies, especially on non-uniform grids with Matsubara points at very high frequencies. However, a cutoff of Matsubara frequencies in the input data, or via the Pick selection criterion, eliminates this information, leading to spectral functions that may have incorrect moments. Imposing constraints on the moments during the interpolation can therefore improve the accuracy of the continued fraction in the Nevanlinna analytic continuation process. The enforcement of moments and the combination of the moment with the interpolation problem is known as the Hamburger Moment Problem [44, 46, 61].

Let us consider a sequence of moments, $b = (h_0, h_1, h_2, \ldots, h_{2N-2})$. The vector $b$ is referred to as the Hankel vector and can be pre-calculated using the equations of motion [62, 63]. Our objective is to determine a non-decreasing measure $\sigma(\omega)$ that satisfies the following equation:

$$h_n = \int_{-\infty}^{\infty} \omega^n d\sigma(\omega). \tag{44}$$

for $n = 0, 1, 2, \ldots, 2N-2$. The spectral function is expressed as $\rho(\omega) = \frac{d\sigma(\omega)}{d\omega} (\geq 0)$. According to the Hamburger-Nevanlinna theorem [61], there is a one-to-one correspondence between the class of solutions $\sigma(\omega)$ and a subset of Nevanlinna functions:

$$f(z) = \int_{-\infty}^{\infty} \frac{d\sigma(\omega)}{\omega - z}. \tag{45}$$

This Nevanlinna function has the following asymptotic form:

$$f(z) = -\frac{h_0}{z} - \frac{h_1}{z^2} - \frac{h_2}{z^3} - \cdots - \frac{h_{2N-2}}{z^{2N-1}} - o\left(\frac{1}{z^{2N-1}}\right), \tag{46}$$

where the domain of $f(z)$ is $\epsilon < \arg z < \pi - \epsilon$ for some $0 < \epsilon < \frac{\pi}{2}$.

The continuation of $f(z)$ is only possible if the Hankel matrix $H_{NN}[b]$, which is defined as follows:

$$H_{kl}[b] = \left(h_{i+j}\right)_{i,j=0}^{i=k-1, j=l-1}, \quad k + l = 2N \tag{47}$$

is considered "proper". The characteristic degrees of the Hankel matrix are defined as $n_1 = \text{rank} \, H_{NN}[b]$ and $n_2 = 2N - n_1$. A Hankel matrix $A$ is considered proper when its leading submatrix, $B = \left(A_{i,j}\right)_{i,j=0}^{i=n_1-1, j=n_1-1}$, of order $n_1 \times n_1$ is non-singular, and thus $n_1 = \text{rank} \, B$ [64]. Note that a non-singular Hankel matrix is proper.

We introduce a polynomial space defined by the kernel of the Hankel matrix, as given by the following equation:

$$\mathcal{A}_l = \left(1, z, z^2, \ldots, z^{l-1}\right) \text{ker}(H_{kl}[b]). \quad k + l = 2N \tag{48}$$

In constructing a solution, we utilize two distinct types of polynomials. Let us denote the first type as $p(z)$ and $q(z)$. When $n_1 = n_2 = N$, the dimension of $\mathcal{A}_{n_1+1}$ is 2 and $p(z)$ and $q(z)$ serve as a basis for this space. However, when $n_1 < n_2$, $\mathcal{A}_{n_1+1}$ has a dimension of 1 and $p(z)$ serves as its basis. Meanwhile, the set $p(z), zp(z), \ldots, z^{n_2-n_1}p(z), q(z)$ forms an orthogonal basis for $\mathcal{A}_{n_2+1}$.

The polynomials $p(z)$ and $q(z)$ are not uniquely defined, but a special pair of canonical polynomials is often utilized for convenience. The expression for $n_1$-th order polynomial is given by

$$
\alpha \det \begin{pmatrix}
h_0 & h_1 & \cdots & h_{n_1} \\
h_1 & h_2 & \cdots & h_{n_1+1} \\
\vdots & \vdots & & \vdots \\
h_{n_1-1} & h_{n_1} & \cdots & h_{2n_1-1} \\
1 & z & \cdots & z^{n_1}
\end{pmatrix},
\tag{49}
$$

where $\alpha$ is a normalization coefficient that ensures that the polynomial is monic. In the case where $n_1 = N$, $h_{2n_1-1}$ is an arbitrary real number [60]. We choose $p(z)$ to be an $n_1$-th order orthogonal polynomial and $q(z)$ to be an $(n_1 - 1)$-th order polynomial. The polynomials can be expressed as:

$$
p(z) = \sum_{n=0}^{n_1} p_n z^n,
\tag{50}
$$

$$
q(z) = \sum_{n=0}^{n_2} q_n z^n.
\tag{51}
$$

Additionally, we define the symmetrizers of $p(z)$ and $q(z)$ as follows:

$$
S(p(z)) = \begin{pmatrix}
p_1 & \cdots & p_{n_1-1} & p_{n_1} \\
\vdots & \ddots & \ddots & 0 \\
p_{n_1-1} & \ddots & \ddots & \vdots \\
p_{n_1} & 0 & \cdots & 0
\end{pmatrix},
\tag{52}
$$

$$
S(q(z)) = \begin{pmatrix}
q_1 & \cdots & q_{n_2-1} & q_{n_2} \\
\vdots & \ddots & \ddots & 0 \\
q_{n_2-1} & \ddots & \ddots & \vdots \\
q_{n_2} & 0 & \cdots & 0
\end{pmatrix}.
\tag{53}
$$

Finally, we introduce another two sets of polynomials, which are the conjugate polynomials of $p(z)$ and $q(z)$:

$$
\gamma(z) = \left(1, z, z^2, \ldots, z^{n_1-1}\right) S(p(z)) \left(h_0, h_1, \ldots, h_{n_1-1}\right)^\top,
\tag{54}
$$

$$
\delta(z) = \left(1, z, z^2, \ldots, z^{n_2-1}\right) S(q(z)) \left(h_0, h_1, \ldots, h_{n_2-1}\right)^\top.
\tag{55}
$$

The solutions to the problem are provided for both the case of a positive definite Hankel matrix ($H_{NN} > 0$) and the case of a semi-positive definite Hankel matrix ($H_{NN} \geq 0$), as follows

(see Theorem 3.6 in Ref. [60]):

$$f(z) = \int_{-\infty}^{\infty} \frac{d\sigma(\omega)}{\omega - z} \tag{56}$$

$$= \begin{cases} -\dfrac{\gamma(z) + \varphi(z)\delta(z)}{p(z) + \varphi(z)q(z)} & (H_{NN} > 0), \\[4mm] -\dfrac{\gamma(z)}{p(z)} & (H_{NN} \geq 0 \text{ and proper}). \end{cases} \tag{57}$$

Here, $\varphi(z)$ represents any Nevanlinna function such that $\varphi(z)/z$ approaches zero as $|z|$ approaches infinity.

These frameworks can be combined with the Schur algorithm by incorporating Nevanlinna analytic continuation. Given the data for $f(z)$ to be interpolated,

$$f(Y_\alpha) = \lambda_\alpha \quad (\alpha = 1, 2, \ldots, M), \tag{58}$$

we modify data by polynomials $p(z), q(z), \gamma(z), \delta(z)$, as follows:

$$\varphi(Y_\alpha) = \tilde{\lambda}_\alpha = -\frac{\gamma(Y_\alpha) + \lambda_\alpha p(Y_\alpha)}{\delta(Y_\alpha) + \lambda_\alpha q(Y_\alpha)} \quad (\alpha = 1, 2, 3, \ldots, M). \tag{59}$$

Since $\varphi(z)$ is a Nevanlinna function, the Schur algorithm interpolates the data in Eq. (59) and gives $\varphi(z)$ and $f(z)$.

# 3 Usage

## 3.1 Installation

Firstly, users need to install Julia (v1.6 or newer) and make sure to add the location of the Julia executable (`julia`) to your PATH environment variable.

Installing the library is straightforward, thanks to Julia's package manager. To start, open Julia using the REPL (read-eval-print loop), which is an interactive command-line interface, and press the ] key to activate the package mode. Then enter the following:

```
pkg> add Nevanlinna
```

Upon successful installation, you'll be able to use our library in a Julia session as follows:

```
julia> using Nevanlinna
```

Alternatively, the libraries can be installed in a shell as follows:

```
$ julia -e 'import Pkg; Pkg.add("Nevanlinna")'
```

This command tells Julia to import the package management system and add (i.e., install) the `Nevanlinna.jl` package. This installation will be performed in the currently active environment in your Julia session.

If you intend to run the sample code provided later in this paper, it will also be necessary to install `SparseIR.jl` [65] for the sparse sampling method [66] based on the intermediate representation [67]. You can do this by adding it in the same way as `Nevanlinna.jl`. In the Julia package mode, simply type the following command:

```
pkg> add SparseIR
```

Alternatively, you can install the package directly from the shell by entering the following command:

```
$ julia -e 'import Pkg; Pkg.add("SparseIR")'
```

## 3.2 Interface

### 3.2.1 REPL or Jupyter notebook

The `Nevanlinna.jl` package can be utilized within either a REPL or Jupyter notebook. First, arrays containing data for the Matsubara Green's function $\mathcal{G}(i\omega_n)$ and the Matsubara frequency $i\omega_n$ are needed. The constructor `NevanlinnaSolver` and `HamburgerNevanlinnaSolver` can be used for the bare Nevanlinna analytic continuation and the Hamburger moment problem combined with Nevanlinna analytic continuation, respectively: For the bare Nevanlinna analytic continuation,

```julia
julia> sol = NevanlinnaSolver(wn, gw, N_real, w_max, eta, sum_rule,
    H_max, iter_tol, lambda)
```

For the Hamburger moment problem,

```julia
julia> sol = HamburgerNevanlinnaSolver(moments, wn, gw, N_real,
    w_max, eta, sum_rule, H_max, iter_tol, lambda)
```

In the above code, `wn` and `gw` are the arrays of $i\omega_n$ and $\mathcal{G}(i\omega_n)$, while `moments` contains the data of moments of $\rho(\omega)$. `N_real` represents the number of mesh points in the real axis and `w_max` represents the energy cutoff of the real axis. `eta` and `sum_rule` describe the broaden parameter $\eta$ and $\int d\omega\, \rho(\omega)$ respectively. `H_max`, `iter_tol`, and `lambda` define the upper cutoff of $H$ in Hardy optimization, the upper bound of iteration, the regularization parameter in Eq. (38) which are hyperparameters used in calculations. The other parameters are summarized in Table 1. The constructor `HamburgerNevanlinnaSolver` requires an additional input array, `moments`. Within the constructors, the optimal values for $N_{\mathrm{opt}}$ and $H_{\mathrm{min}}$ are calculated automatically. The Hardy optimization can then be performed by executing the `solve!` function, as shown below:

```julia
julia> solve!(sol)
```

### 3.2.2 CLI (command line interface)

For the convenience of the user, the `Nevanlinna.jl` package also offers a command-line interface. Upon installation of `Nevanlinna.jl` via Julia, an executable file named `nevanlinna` is automatically created in the `~/.julia/bin` directory. Assuming the path to the executable is already included in your system's PATH, the following commands can be executed:

```
$ nevanlinna bare inputpath parampath outputpath
$ nevanlinna hamburger inputpath momentpath parampath outputpath
```

The first argument determines the calculation mode, which should be either `bare` (for bare Nevanlinna analytic continuation) or `hamburger` (for the Hamburger moment problem).

The second argument, `inputpath`, denotes the path to the data file that contains the Matsubara frequency $i\omega_n$ and the Matsubara Green's function $\mathcal{G}(i\omega_n)$. This file should contain $\omega_n$, $\Re(\mathcal{G}(i\omega_n))$, and $\Im(\mathcal{G}(i\omega_n))$ data within the first, second, and third columns, respectively.

The third argument, `parampath`, is the path to the input parameter file in TOML format. A template TOML file is provided in the associated GitHub repository. For the Hamburger moment problem mode, an additional third argument, `momentpath`, specifies the path to the moment data file, which should have the moment data in the first column.

The final argument, `outputpath`, is the path to the output data file, where the frequency $\omega$ on the real axis and the resulting analytic continuation data $\rho(\omega)$ are stored in the first and second columns, respectively.

Table 1: Arguments of constructors of `NevanlinnaSolver` and `HamburgerNevanlinnaSolver`. The first argument, `moments`, is needed only for `HamburgerNevanlinnaSolver`.

| Variable | Type | Description |
|---|---|---|
| `moments` | `Vector{Complex{T}}` | Array of $h_n$ |
| | | Only for `HamburgerNevanlinnaSolver` |
| `wn` | `Vector{Complex{T}}` | Array of $i\omega_n$ |
| `gw` | `Vector{Complex{T}}` | Array of $\mathcal{G}(i\omega_n)$ |
| `N_real` | `Int64` | The number of mesh in the real axis |
| `w_max` | `Float64` | Energy cutoff of the real axis |
| `eta` | `Float64` | Broaden parameter $\eta$ |
| `sum_rule` | `Float64` | $\int d\omega\, \rho(\omega)$ |
| `H_max` | `Int64` | Upper cutoff of $H$ |
| `iter_tol` | `Int64` | Upper bound of iteration |
| `lambda` | `Float64` | Regularization parameter $\lambda$ |
| `verbose` | `Bool` | Verbose option |
| | | (Default: `false`) |
| `pick_check` | `Bool` | Causality check option |
| | | (Default: `true`) |
| `optimization` | `Bool` | Hardy optimization option |
| | | (Default: `true`) |
| `ini_iter_tol` | `Int64` | Upper bound of iteration for $H_{\min}$ |
| | | (Default: 500) |
| `mesh` | `Symbol` | Mesh on the real axis option |
| | | (Default: `:linear`) |

### 3.3  Examples

To illustrate the capabilities of our code, we present a numerical analytic continuation for several models, which include a $\delta$-function, a Gaussian, a Lorentzian, a two-peak, a Kondo resonance, and a Hubbard gap model. Jupyter notebooks, which can be used to execute these examples, are provided in the `notebooks` directory of our repository. The three of these models were previously analyzed in Ref. [44]. The exact spectral functions for these models are given by the following equations:

$$\rho^{\delta\text{-function}}(\omega) = 0.3\,\delta(\omega-1) + 0.5\,\delta(\omega+3) + 0.2\,\delta(\omega-4.5),$$
$$\rho^{\text{Gaussian}}(\omega) = g(\omega,0,1),$$
$$\rho^{\text{Lorentzian}}(\omega) = l(\omega,0,1),$$
$$\rho^{\text{two peak}}(\omega) = 0.8\,g(\omega,-1,1.0) + 0.2\,g(\omega,3,0.7),$$
$$\rho^{\text{Kondo resonance}}(\omega) = 0.45\,g(\omega,-2.5,0.7) + 0.1\,g(\omega,0,0.1) + 0.45\,g(\omega,2.5,0.7)$$
$$\rho^{\text{Hubbard gap}}(\omega) = 0.5\,g(\omega,-1.9,0.5) + 0.5\,g(\omega,1.9,0.5) \tag{60}$$

where

$$g(\omega,\mu,\sigma) = \frac{1}{\sqrt{2\pi}\sigma}\exp\left\{-\frac{(x-\mu)^2}{2\sigma^2}\right\},$$
$$l(\omega,\mu,\gamma) = \frac{1}{\pi}\frac{\gamma}{(\omega-\gamma)^2 + \gamma^2}. \tag{61}$$

We prepare double precision input data $\mathcal{G}(i\omega_n)$ on a sparse sampling grid of Matsubara frequencies, i.e., the intermediate-representation [67] grid for $\beta = 100$ [66], generated by using SparseIR.jl [65]. The code can be found in Fig. B.2. After the analytic continuation is performed, the output data can be accessed through `sol.reals`. We evaluate the continued results on $\omega + 0.001i$ and show them in Fig. 1. Except for the $\delta$-function model, the continued result shows artificial oscillations around the exact spectral function in the absence of Hardy optimization. However, by the Hardy optimization implemented in our code, these oscillations are effectively removed and the continued spectral function is in good agreement with the exact function in all cases.

To demonstrate the significance of utilizing multiple precision arithmetic in the Schur algorithm, we compare the results obtained with 64-bit arithmetic and 128-bit arithmetic. The optimized result is shown in Fig. 2. The result obtained with 64-bit arithmetic is incorrect, as the small peak is not properly restored and there is finite spectral weight in the high-$\omega$ region. This indicates that the rounding error in the Schur algorithm can significantly affect the continued result. Hence, employing multiple precision arithmetic is essential to ensure that rounding errors remain negligible throughout the computations.

The case in which the spectral function displays a large gap around the origin is known to be challenging. The kernel of analytic continuation implies that information about the spectral function may be lost in the Matsubara Green's function [67]. Consequently, the Matsubara Green's function in such cases exhibits a lower tolerance for noise. Computations at high temperatures yield a *qualitatively correct* solution. Figure 3 shows the results for this case. The positions and weights of the peaks are reconstructed; however, some small oscillations remain. Implementing a more robust algorithm for Hardy optimization will enhance the performance of our code.

In computations at low temperatures, the Matsubara frequencies are close to each other in the complex $\omega$-plane, making it difficult to access high-frequency behavior that may have been truncated by Pick selection. Including information about the moments can improve the results

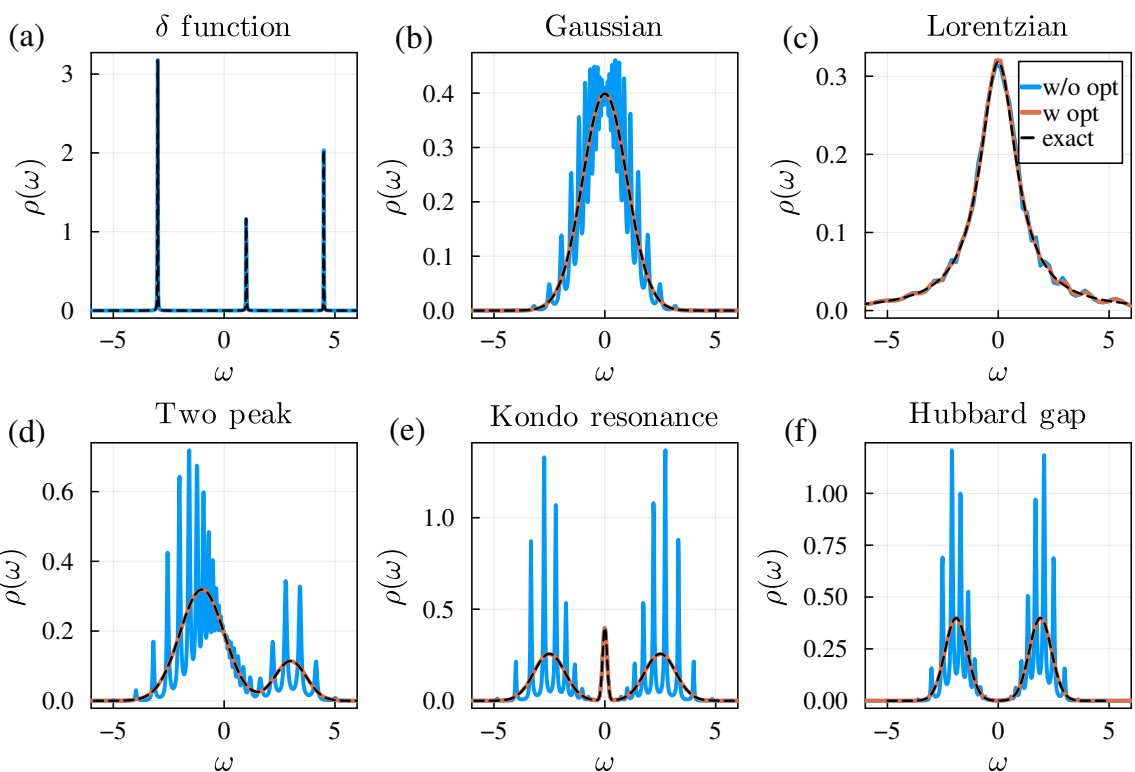

Figure 1: Results of (a) $\delta$-function, (b) Gaussian, (c) Lorentzian, (d) two-peak, (e) Kondo-resonance, (f) Hubbard-gap models with and without optimization in `Nevanlinna.jl`. These results were obtained for $\beta = 100$ and $\eta = 0.001$ The exact spectral functions consist of $\delta$-function, Gaussian peaks, or Lorentzian peaks [Eq. (60)].

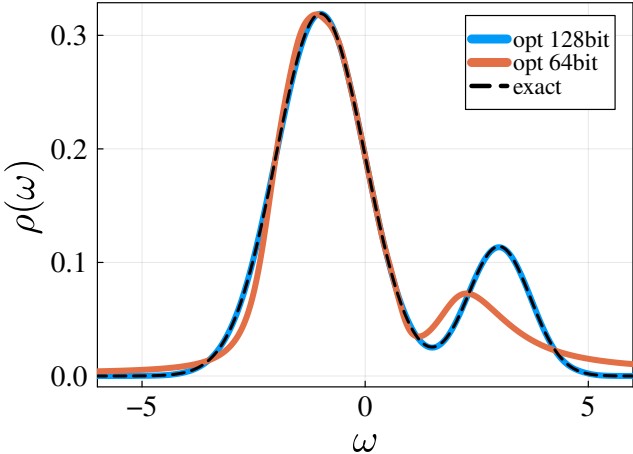

Figure 2: Results of the two-peak model obtained by 64-bit and 128-bit arithmetic. The spectral function is the same as Fig. 1(d).

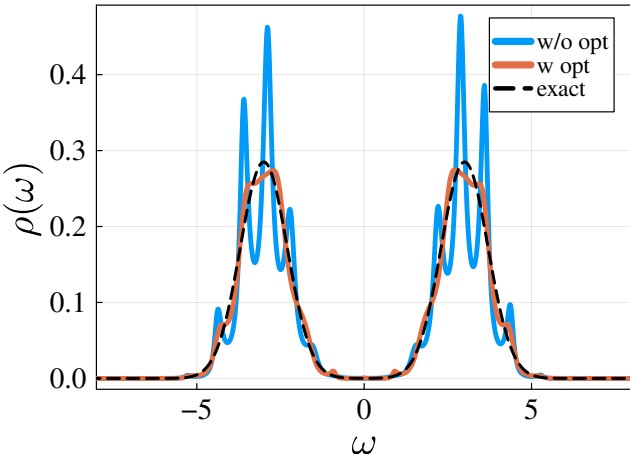

Figure 3: Results of the large Hubbard gap model for $\beta = 10$ and $\eta = 0.01$. The spectral function is $0.5 * g(\omega, -3.0, 0.7) + 0.5 * g(\omega, 3.0, 0.7)$

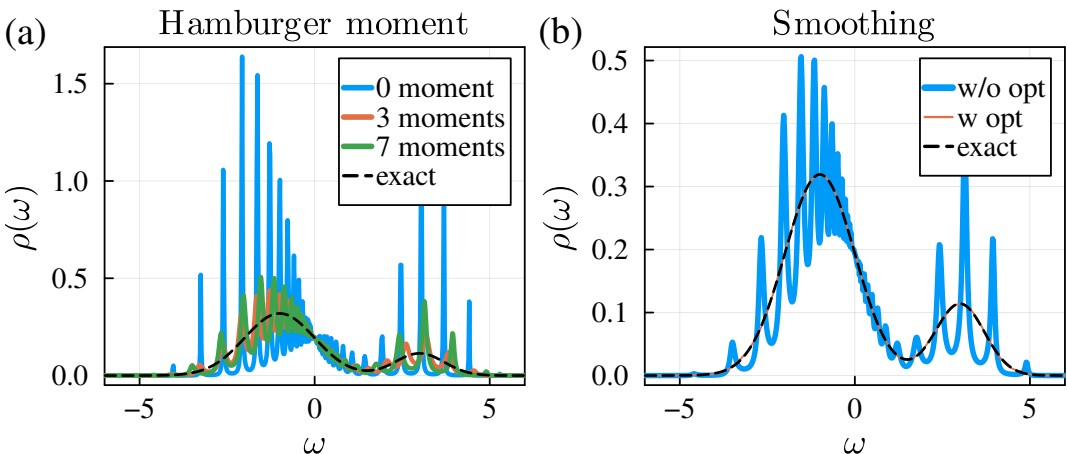

Figure 4: (a) Results of the two-peak model for $\beta = 1000$ and $\eta = 0.0001$. We imposed constraints on the first 0, 3, and 7 moments, respectively. (b) Results with smoothing and constraints on the first seven moments. The spectral function is the same as Fig. 1(d).

in these situations. Figure 4(a) illustrates the influence of the use of moment information on the outcomes. The incorporation of additional information leads to a reduction of artificial oscillations. This augmentation stabilizes the numerical computation during Hardy optimization. The Hardy optimization still works efficiently even in the case of the Hamburger moment problem (Fig. 4(b)). The inclusion of moments is beneficial in low-temperature calculations or situations where input data is limited.

## 4   Conclusion

In this paper, we introduced the Julia library `Nevanlinna.jl`. We provided an overview of the analytic structure of the Green's function, Schur algorithm, Pick criterion, Hardy optimization, and Hamburger moment problem. The Matsubara and retarded Green's function on the upper half-plane are classified into the Nevanlinna function. The Schur algorithm effectively interpolates and constructs a Nevanlinna function, ensuring causality automatically. The Pick

criterion serves as the mathematical base for the existence of Nevanlinna interpolants. We implemented the Hardy optimization using efficient automatic differentiation. The Hamburger moment problem enables analytic continuation with constraints on the moments of a spectral function. We demonstrated the usage of our code with various examples such as $\delta$-function, Gaussian, Lorentzian, a two-peak, a Kondo resonance, and Hubbard gap models.

The installation of our code is extraordinarily easy using the Julia package manager. Furthermore, multiple precision arithmetic is already implemented. Thus, there is no obstacle, such as compiling the code or installing an external library manually, and users can readily try our code.

Finally, we discuss some remaining technical issues and further extensions to be addressed. In some cases, like the large Hubbard gap structure, our Hardy optimization algorithm may fail to find the optimal solution $\theta_{M+1}(z)$. However, the Pick criterion guarantees the existence of the *true* undetermined function $\theta_{M+1}(z)$. Therefore, further investigation into the optimization algorithm will improve the range of applications of Nevanlinna analytic continuation. Although our code currently employs Cholesky decomposition to verify the semi-positive definiteness of Pick or Hankel matrices, it is well-known that robust criteria and efficient algorithms exist to confirm the positive definiteness of given matrices [56]. Implementing this algorithm into our code is a direction for future work. The extension for the matrix-valued Green's function is also an interesting topic. While this topic is resolved for spectral functions like the $\delta$-function [45], broadened cases have not been investigated yet. In addition, further expansion of Nevanlinna analytic continuation to self-energy [68] or anomalous Green's function [69] is crucial for wide-range applications of many-body physics.

# Acknowledgements

The authors are grateful to T. Koretsune, S. Namerikawa, and F. Kakizawa for fruitful discussions.

**Funding information** K.N. was supported by JSPS KAKENHI (Grants No. JP21J23007) and Research Grants, 2022 of WISE Program, MEXT. H.S. was supported by JSPS KAKENHI Grants No. 18H01158, No. 21H01041, and No. 21H01003, and JST PRESTO Grant No. JP-MJPR2012, Japan. E.G. was supported by the National Science Foundation under Grant No. NSF DMR 2001465.

The code is available under the MIT license at https://github.com/SpM-lab/Nevanlinna.jl

# A Structure of code

## A.1 Processing flow

The function `calc_opt_N_imag` calculates the *optimal* cutoff number `opt_N_imag`, aiming to preserve causality, as described in Sec. 2.2.3. Then, with the calculated `opt_N_imag`, `calc_phis` calculate $\phi_\alpha$ as described in Sec. 2.2.4. Following this, `calc_abcd` evaluates the functions $a(z)$, $b(z)$, $c(z)$, and $d(z)$ at $z = \omega + i\eta$ using the Schur algorithm. Finally, optimal `H_min` is evaluated by `calc_H_min`, and the Hardy optimization is executed. The flowchart of our procedure is shown in Fig. A.1, and a summary of the functions used in the procedure is provided in Table A.1.

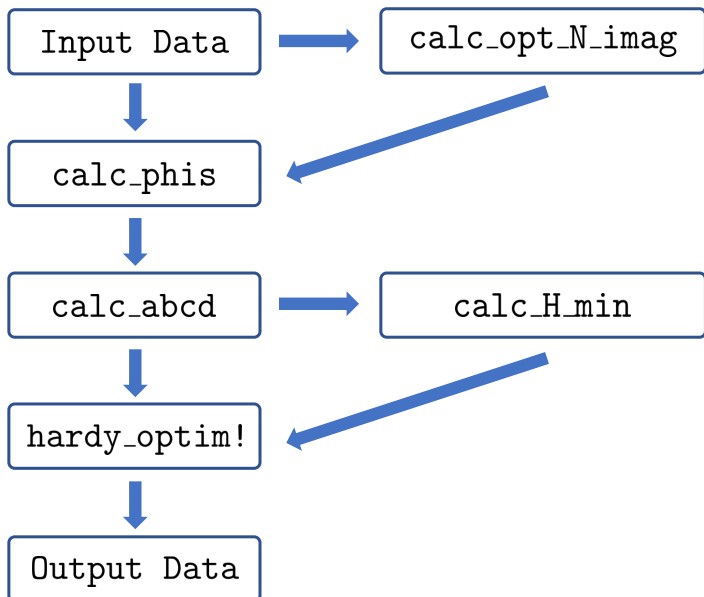

Figure A.1: Flowchart of `Nevanlinna.jl`

Table A.1: Functions in processing flow

| Variable | Described section |
|---|---|
| calc_opt_N_imag | Sec. 2.2.3 |
| calc_phis | Sec. 2.2.4 |
| calc_abcd | Sec. 2.2.4 |
| calc_H_min | Sec. 2.2.5 |

## A.2   Data struct

We have defined two struct types for input and output data. The struct `ImagDomainData` is used to store input data. In Table A.2, the member variables of `ImagDoaminData` are summarized. `freq` and `val` store $i\omega_n$ and $h_i(-\mathcal{G}(i\omega_n))$, respectively, while `N_imag` represents the dimenson of `freq` and `val`. Similarly, the `RealDomainData` struct is used to store output data. The member variables of the `RealDomainData` struct are summarized in Table A.3. The variables `freq` and `val` store $\omega + i\eta$ and $-G^R(\omega + i\eta)$, respectively, `N_real` represents the dimensons of both `freq` and `val`, `omega_max` represents the energy cutoff of the real axis, `eta` is the broaden parameter, and `sum_rule` corresponds to the value of $\int d\omega \, \rho(\omega)$.

Table A.2: Members of `ImagDomainData`

| Variable | Type | Description |
|---|---|---|
| N_imag | Int64 | Dimension of `freq` and `val` |
| freq | Vector{Complex{T}} | $i\omega_n$ |
| val | Vector{Complex{T}} | $h_i(-\mathcal{G}(i\omega_n))$ |

Table A.3: Members of `RealDomainData`

| Variable | Type | Description |
|---|---|---|
| N_real | Int64 | Dimension of `freq` and `val` |
| w_max | Float64 | Energy cutoff of the real axis |
| eta | Float64 | Broaden parameter $\eta$ |
| sum_rule | Float64 | $\int d\omega \, \rho(\omega)$ |
| freq | Vector{Complex{T}} | $\omega + i\eta$ |
| val | Vector{Complex{T}} | $-G^R(\omega + i\eta)$ |

## A.3   Solver struct

We have defined solver structs for the Nevanlinna analytic continuation and the Hamburger moment problem combined with Nevanlinna analytic continuation. The member variables of these structs are summarized in Table A.4 and Table A.5, respectively. The constructor executes the process from `calc_opt_N_imag` to `calc_H_min` in the flowchart in Fig. A.1. The function `solve!` executes the Hardy optimization step and `RealDomainData` in the `NevanlinnaSolver` contains output data.

## B   Example code

In this section, we present an example code using `Nevanlinna.jl` for the two-peak model. The corresponding results are illustrated in Fig.1(d). Users can apply our code to different spectral functions by modifying the definition of `rho(omega)`.

## References

[1] A. A. Abrikosov, L. P. Gorkov and I. E. Dzyaloshinski, *Methods of quantum field theory in statistical physics,* Courier Corporation (2012).

Table A.4: Members of NevanlinnaSolver

| Variable | Type | Description |
|---|---|---|
| imags | ImagDomainData{T} | Imaginary domain data |
| reals | RealDomainData{T} | Real domain data |
| phis | Vector{Complex{T}} | $\phi_i$ |
| abcd | Array{Complex{T},3} | $a(z), b(z), c(z),$ and $d(z)$ |
| H_max | Int64 | Upper cutoff of $H$ |
| H_min | Int64 | Lower cutoff of $H$ |
| H | Int64 | Current value of $H$ |
| ab_coeff | Vector{Complex{T}} | Current solution for $a_k, b_k$ |
| hardy_matrix | Array{Complex{T},2} | Hardy matrix for $H$ |
| iter_tol | Int64 | Upper bound of iteration |
| lambda | Float64 | Regularization parameter |
| ini_iter_tol | Int64 | upper bound of iteration for $H_{\min}$ |
| verbose | Bool | Verbose option |

Table A.5: Members of HamburgerNevanlinnaSolver

| Variable | Type | Description |
|---|---|---|
| moments | Vector{Complex{T}} | $h_n$ |
| N_moments_ | Int64 | Dimension of moments |
| N | Int64 | (N_moments_+1)/2 |
| $n_1$ | Int64 | rank $H_{NN}[b]$ |
| $n_1$ | Int64 | $2N - n_1$ |
| isPSD | Bool | Whether is $H_{NN}[b]$ positive semi-definite or not |
| isProper | Bool | Whether is $H_{NN}[b]$ proper or not |
| isProper | Bool | Whether is $H_{NN}[b]$ singular or not |
| isDegenerate | Bool | Whether is $H_{NN}[b]$ degenerate or not |
| p | Vector{Complex{T}} | $p_i$ |
| q | Vector{Complex{T}} | $q_i$ |
| gamma | Vector{Complex{T}} | $\gamma_i$ |
| delta | Vector{Complex{T}} | $\delta_i$ |
| hankel | Array{Complex{T},2} | Hankel matrix $H_{NN}[b]$ |
| mat_real_omega | Array{Complex{T},2} | Matrix of $\omega^n$ |
| val | Vector{Complex{T}} | $f(z)$ |
| nev_st | NevanlinnaSolver{T} | NevanlinnaSolver for $\phi(z)$ |
| verbose | Bool | Verbose option |

```julia
1    #load package
2    using Nevanlinna
3    using LinearAlgebra
4    using SparseIR
5
6    #set work data Type
7    T = BigFloat
8    setprecision(128)
9
10   #define spectral function
11   gaussian(x, mu, sigma) = exp(-0.5*((x-mu)/sigma)^2)/(sqrt(2*π)*
         sigma)
12   rho(omega) = 0.8*gaussian(omega, -1.0, 1.0) + 0.2*gaussian(omega,
         3, 0.7)
13
14   function generate_input_data(rho::Function, beta::Float64)
15       lambda = 1e+4
16       wmax = lambda/beta
17       basis = FiniteTempBasisSet(beta, wmax, 1e-15)
18
19       rhol = [overlap(basis.basis_f.v[l], rho) for l in 1:length(
           basis.basis_f)]
20       gl = - basis.basis_f.s .* rhol
21       gw = evaluate(basis.smpl_wn_f, gl)
22
23       hnw = length(basis.smpl_wn_f.sampling_points)÷2
24
25       input_smpl = Array{Complex{T}}(undef, hnw)
26       input_gw   = Array{Complex{T}}(undef, hnw)
27       for i in 1:hnw
28           input_smpl[i]= SparseIR.valueim(basis.smpl_wn_f.
               sampling_points[hnw+i], beta)
29           input_gw[i]  = gw[hnw+i]
30       end
31       return input_smpl, input_gw
32   end
33
34   beta = 100. #inverse temperature
35   input_smpl, input_gw = generate_input_data(rho, beta)
36
37   N_real    = 1000  #dimension of the array of output
38   omega_max = 10.0  #energy cutoff of the real axis
39   eta       = 0.001 #broaden parameter
40   sum_rule  = 1.0   #sum rule
41   H_max     = 50    #cutoff of Hardy basis
42   lambda    = 1e-4  #regularization parameter
43   iter_tol  = 1000  #upper bound of iteration
44
45   #construct solver struct
46   sol = NevanlinnaSolver(input_smpl, input_gw, N_real, omega_max, eta
         , sum_rule, H_max, iter_tol, lambda, verbose=true)
47
48   #execute optimize
49   solve!(sol)
```

Figure B.2: Example code for the two-peak model. This code is available at https://github.com/SpM-lab/Nevanlinna.jl/notebooks/two_peak.ipynb.

[2] A. L. Fetter and J. D. Walecka, *Quantum theory of many-particle systems*, Courier Corporation (2012).

[3] A. M. Zagoskin, *Quantum theory of many-body systems*, vol. 174, Springer, doi:https://doi.org/10.1007/978-3-319-07049-0 (1998).

[4] R. D. Mattuck, *A guide to Feynman diagrams in the many-body problem*, Courier Corporation (1992).

[5] G. D. Mahan, *Many-particle physics*, Springer Science & Business Media, doi:https://doi.org/10.1007/978-1-4757-5714-9 (2013).

[6] J. W. Negele, *Quantum many-particle systems*, CRC Press, doi:https://doi.org/10.1201/9780429497926 (2018).

[7] A. Altland and B. D. Simons, *Condensed matter field theory*, Cambridge university press, doi:https://doi.org/10.1017/CBO9780511789984 (2010).

[8] P. W. Anderson, *Absence of diffusion in certain random lattices*, Phys. Rev. **109**, 1492 (1958), doi:10.1103/PhysRev.109.1492.

[9] P. Nozieres, *Theory of interacting Fermi systems*, CRC Press, doi:https://doi.org/10.1201/9780429495724 (2018).

[10] D. Pines, *Theory of Quantum Liquids: Normal Fermi Liquids*, CRC Press, doi:https://doi.org/10.4324/9780429492662 (2018).

[11] S. Onari, Y. Yamakawa and H. Kontani, *Sign-reversing orbital polarization in the nematic phase of fese due to the $C_2$ symmetry breaking in the self-energy*, Phys. Rev. Lett. **116**, 227001 (2016), doi:10.1103/PhysRevLett.116.227001.

[12] R. Tazai, S. Matsubara, Y. Yamakawa, S. Onari and H. Kontani, *Rigorous formalism for unconventional symmetry breaking in fermi liquid theory and its application to nematicity in fese*, Phys. Rev. B **107**, 035137 (2023), doi:10.1103/PhysRevB.107.035137.

[13] H. Kontani, R. Tazai, Y. Yamakawa and S. Onari, *Unconventional density waves and superconductivities in fe-based superconductors and other strongly correlated electron systems*, Advances in Physics **0**(0), 1 (2023), doi:10.1080/00018732.2022.2144590, https://doi.org/10.1080/00018732.2022.2144590.

[14] T. Moriya and K. Ueda, *Spin fluctuations and high temperature superconductivity*, Advances in Physics **49**(5), 555 (2000), doi:10.1080/000187300412248, https://doi.org/10.1080/000187300412248.

[15] Y. Yanase, T. Jujo, T. Nomura, H. Ikeda, T. Hotta and K. Yamada, *Theory of superconductivity in strongly correlated electron systems*, Physics Reports **387**(1), 1 (2003), doi:https://doi.org/10.1016/j.physrep.2003.07.002.

[16] A. Georges, G. Kotliar, W. Krauth and M. J. Rozenberg, *Dynamical mean-field theory of strongly correlated fermion systems and the limit of infinite dimensions*, Rev. Mod. Phys. **68**, 13 (1996), doi:10.1103/RevModPhys.68.13.

[17] J. E. Hirsch and R. M. Fye, *Monte carlo method for magnetic impurities in metals*, Phys. Rev. Lett. **56**, 2521 (1986), doi:10.1103/PhysRevLett.56.2521.

[18] A. N. Rubtsov, V. V. Savkin and A. I. Lichtenstein, *Continuous-time quantum monte carlo method for fermions*, Phys. Rev. B **72**, 035122 (2005), doi:10.1103/PhysRevB.72.035122.

[19] P. Werner, A. Comanac, L. de' Medici, M. Troyer and A. J. Millis, *Continuous-time solver for quantum impurity models*, Phys. Rev. Lett. **97**, 076405 (2006), doi:10.1103/PhysRevLett.97.076405.

[20] P. Werner and A. J. Millis, *Hybridization expansion impurity solver: General formulation and application to kondo lattice and two-orbital models*, Phys. Rev. B **74**, 155107 (2006), doi:10.1103/PhysRevB.74.155107.

[21] E. Gull, A. J. Millis, A. I. Lichtenstein, A. N. Rubtsov, M. Troyer and P. Werner, *Continuous-time monte carlo methods for quantum impurity models*, Rev. Mod. Phys. **83**, 349 (2011), doi:10.1103/RevModPhys.83.349.

[22] T. DeGrand and C. DeTar, *Lattice Methods for Quantum Chromodynamics*, WORLD SCIENTIFIC, doi:10.1142/6065 (2006), https://www.worldscientific.com/doi/pdf/10.1142/6065.

[23] C. Gattringer and C. Lang, *Quantum chromodynamics on the lattice: an introductory presentation*, vol. 788, Springer Science & Business Media, doi:https://doi.org/10.1007/978-3-642-01850-3 (2009).

[24] H. J. Rothe, *Lattice Gauge Theories : An Introduction (Fourth Edition)*, vol. 43, World Scientific Publishing Company, ISBN 978-981-4365-87-1, 978-981-4365-85-7, doi:10.1142/8229 (2012).

[25] A. Filinov, *Correlation effects and collective excitations in bosonic bilayers: Role of quantum statistics, superfluidity, and the dimerization transition*, Phys. Rev. A **94**, 013603 (2016), doi:10.1103/PhysRevA.94.013603.

[26] K. Nogaki and H. Shinaoka, *Bosonic nevanlinna analytic continuation*, Journal of the Physical Society of Japan **92**(3), 035001 (2023), doi:10.7566/JPSJ.92.035001.

[27] M. Boninsegni and D. M. Ceperley, *Density fluctuations in liquid4he. path integrals and maximum entropy*, Journal of Low Temperature Physics **104**(5), 339 (1996), doi:10.1007/BF00751861.

[28] E. Vitali, M. Rossi, L. Reatto and D. E. Galli, *Ab initio low-energy dynamics of superfluid and solid $^4He$*, Phys. Rev. B **82**, 174510 (2010), doi:10.1103/PhysRevB.82.174510.

[29] S. Saccani, S. Moroni and M. Boninsegni, *Excitation spectrum of a supersolid*, Phys. Rev. Lett. **108**, 175301 (2012), doi:10.1103/PhysRevLett.108.175301.

[30] T. Dornheim, S. Groth, J. Vorberger and M. Bonitz, *Ab initio path integral monte carlo results for the dynamic structure factor of correlated electrons: From the electron liquid to warm dense matter*, Phys. Rev. Lett. **121**, 255001 (2018), doi:10.1103/PhysRevLett.121.255001.

[31] G. A. Baker, G. A. Baker Jr, P. Graves-Morris and S. S. Baker, *Pade Approximants: Encyclopedia of Mathematics and It's Applications, Vol. 59 George A. Baker, Jr., Peter Graves-Morris*, vol. 59, Cambridge University Press, doi:https://doi.org/10.1017/CBO9780511530074 (1996).

[32] R. K. Bryan, *Maximum entropy analysis of oversampled data problems*, European Biophysics Journal **18**(3), 165 (1990), doi:10.1007/BF02427376.

[33] M. Jarrell and J. Gubernatis, *Bayesian inference and the analytic continuation of imaginary-time quantum monte carlo data*, Physics Reports **269**(3), 133 (1996), doi:https://doi.org/10.1016/0370-1573(95)00074-7.

[34] A. W. Sandvik, *Stochastic method for analytic continuation of quantum monte carlo data*, Phys. Rev. B **57**, 10287 (1998), doi:10.1103/PhysRevB.57.10287.

[35] A. S. Mishchenko, N. V. Prokof'ev, A. Sakamoto and B. V. Svistunov, *Diagrammatic quantum monte carlo study of the fröhlich polaron*, Phys. Rev. B **62**, 6317 (2000), doi:10.1103/PhysRevB.62.6317.

[36] K. Vafayi and O. Gunnarsson, *Analytical continuation of spectral data from imaginary time axis to real frequency axis using statistical sampling*, Phys. Rev. B **76**, 035115 (2007), doi:10.1103/PhysRevB.76.035115.

[37] S. Fuchs, M. Jarrell and T. Pruschke, *Application of bayesian inference to stochastic analytic continuation*, Journal of Physics: Conference Series **200**(1), 012041 (2010), doi:10.1088/1742-6596/200/1/012041.

[38] O. Goulko, A. S. Mishchenko, L. Pollet, N. Prokof'ev and B. Svistunov, *Numerical analytic continuation: Answers to well-posed questions*, Phys. Rev. B **95**, 014102 (2017), doi:10.1103/PhysRevB.95.014102.

[39] H. Yoon, J.-H. Sim and M. J. Han, *Analytic continuation via domain knowledge free machine learning*, Phys. Rev. B **98**, 245101 (2018), doi:10.1103/PhysRevB.98.245101.

[40] J. Otsuki, M. Ohzeki, H. Shinaoka and K. Yoshimi, *Sparse modeling approach to analytical continuation of imaginary-time quantum monte carlo data*, Phys. Rev. E **95**, 061302 (2017), doi:10.1103/PhysRevE.95.061302.

[41] J. Otsuki, M. Ohzeki, H. Shinaoka and K. Yoshimi, *Sparse modeling in quantum many-body problems*, Journal of the Physical Society of Japan **89**(1), 012001 (2020), doi:10.7566/JPSJ.89.012001.

[42] L. Ying, *Analytic continuation from limited noisy matsubara data*, Journal of Computational Physics **469**, 111549 (2022), doi:10.1016/j.jcp.2022.111549.

[43] Z. Huang, E. Gull and L. Lin, *Robust analytic continuation of green's functions via projection, pole estimation, and semidefinite relaxation*, doi:10.48550/ARXIV.2210.04187 (2022).

[44] J. Fei, C.-N. Yeh and E. Gull, *Nevanlinna analytical continuation*, Phys. Rev. Lett. **126**, 056402 (2021), doi:10.1103/PhysRevLett.126.056402.

[45] J. Fei, C.-N. Yeh, D. Zgid and E. Gull, *Analytical continuation of matrix-valued functions: Carathéodory formalism*, Phys. Rev. B **104**, 165111 (2021), doi:10.1103/PhysRevB.104.165111.

[46] J. Fei, *A probe into propagators*, https://dx.doi.org/10.7302/1312, doi:10.7302/1312 (2021).

[47] T. Matsubara, *A New Approach to Quantum-Statistical Mechanics*, Progress of Theoretical Physics **14**(4), 351 (1955), doi:10.1143/PTP.14.351.

[48] H. Umezawa and S. Kamefuchi, *The Vacuum in Quantum Electrodynamics*, Progress of Theoretical Physics **6**(4), 543 (1951), doi:10.1143/ptp/6.4.543.

[49] G. Källén, *On the definition of the renormalization constants in quantum electrodynamics*, Helvetica Physica Acta **25**(IV), 417 (1952), doi:10.5169/seals-112316.

[50] M. Gell-Mann and F. E. Low, *Quantum electrodynamics at small distances*, Phys. Rev. **95**, 1300 (1954), doi:10.1103/PhysRev.95.1300.

[51] H. Lehmann, *Über eigenschaften von ausbreitungsfunktionen und renormierungskonstanten quantisierter felder*, Il Nuovo Cimento (1943-1954) **11**(4), 342 (1954), doi:10.1007/BF02783624.

[52] J. Schur, *Über potenzreihen, die im innern des einheitskreises beschränkt sind.*, Journal für die reine und angewandte Mathematik (Crelles Journal) **1918**(148), 122 (1918), doi:doi:10.1515/crll.1918.148.122.

[53] V. M. Adamyan, J. Alcober and I. M. Tkachenko, *Reconstruction of distributions by their moments and local constraints*, Applied Mathematics Research eXpress **2003**(2), 33 (2003), doi:10.1155/S1687120003212028, https://academic.oup.com/amrx/article-pdf/2003/2/33/6920279/2003-2-33.pdf.

[54] G. Pick, *Über die beschränkungen analytischer funktionen durch vorgegebene funktionswerte*, Mathematische Annalen **78**(1), 270 (1917), doi:10.1007/BF01457103.

[55] P. Khargonekar and A. Tannenbaum, *Non-euclidian metrics and the robust stabilization of systems with parameter uncertainty*, IEEE Transactions on Automatic Control **30**(10), 1005 (1985), doi:10.1109/TAC.1985.1103805.

[56] S. M. Rump, *Verification of positive definiteness*, BIT Numerical Mathematics **46**(2), 433 (2006), doi:10.1007/s10543-006-0056-1.

[57] M. Rosenblum and J. Rovnyak, *Topics in Hardy classes and univalent functions*, Springer Science & Business Media, doi:https://doi.org/10.1007/978-3-0348-8520-1 (1994).

[58] M. Innes, *Don't unroll adjoint: Differentiating ssa-form programs*, CoRR **abs/1810.07951** (2018), 1810.07951.

[59] P. K. Mogensen and A. N. Riseth, *Optim: A mathematical optimization package for Julia*, Journal of Open Source Software **3**(24), 615 (2018), doi:10.21105/joss.00615.

[60] G. ning Chen, *The general rational interpolation problem and its connection with the nevanlinna-pick interpolation and power moment problem*, Linear Algebra and its Applications **273**(1), 83 (1998), doi:https://doi.org/10.1016/S0024-3795(97)00346-7.

[61] N. I. Akhiezer, *The Classical Moment Problem and Some Related Questions in Analysis*, Society for Industrial and Applied Mathematics, Philadelphia, PA, doi:10.1137/1.9781611976397 (2020), https://epubs.siam.org/doi/pdf/10.1137/1.9781611976397.

[62] A.-B. Comanac, *Dynamical mean field theory of correlated electron systems: New algorithms and applications to local observables*, Ph.D. thesis, Columbia University, New York (2007).

[63] E. Gull, *Continuous-Time Quantum Monte Carlo Algorithms for Fermions*, Doctoral thesis, ETH Zurich, Zürich, doi:10.3929/ethz-a-005722583 (2008).

[64] M. Fiedler, *Quasidirect decompositions of hankel and toeplitz matrices*, Linear Algebra and its Applications **61**, 155 (1984), doi:https://doi.org/10.1016/0024-3795(84)90028-4.

[65] M. Wallerberger, S. Badr, S. Hoshino, S. Huber, F. Kakizawa, T. Koretsune, Y. Nagai, K. Nogaki, T. Nomoto, H. Mori, J. Otsuki, S. Ozaki *et al.*, *sparse-ir: Optimal compression and sparse sampling of many-body propagators*, SoftwareX **21**, 101266 (2023), doi:https://doi.org/10.1016/j.softx.2022.101266.

[66] J. Li, M. Wallerberger, N. Chikano, C.-N. Yeh, E. Gull and H. Shinaoka, *Sparse sampling approach to efficient ab initio calculations at finite temperature*, Phys. Rev. B **101**(3), 035144 (2020), doi:10.1103/physrevb.101.035144.

[67] H. Shinaoka, J. Otsuki, M. Ohzeki and K. Yoshimi, *Compressing green's function using intermediate representation between imaginary-time and real-frequency domains*, Phys. Rev. B **96**, 035147 (2017), doi:10.1103/PhysRevB.96.035147.

[68] X. Wang, E. Gull, L. de' Medici, M. Capone and A. J. Millis, *Antiferromagnetism and the gap of a mott insulator: Results from analytic continuation of the self-energy*, Phys. Rev. B **80**, 045101 (2009), doi:10.1103/PhysRevB.80.045101.

[69] E. Gull and A. J. Millis, *Quasiparticle properties of the superconducting state of the two-dimensional hubbard model*, Phys. Rev. B **91**, 085116 (2015), doi:10.1103/PhysRevB.91.085116.