# Peer review of "Nevanlinna.jl: A Julia implementation of Nevanlinna analytic continuation"

_SciPost Physics Codebases, doi:SciPost Phys. Codebases 19-r1.0 (2023) , SciPost Phys. Codebases 19 (2023)_

## Round 1 · Referee Report · Anonymous · 2023-4-27

Strengths
- The paper and the software address an issue that has plagued the quantum Monte Carlo community for years and the authors have applied theory that exploits the analytic structure of the Matsubara Green'S function to improve the state of affairs.
- All the work and the obtained experiences have been made available to the public in a software package.
Weaknesses
- Nothing major, the Installation section assumes previous knowledge of the user about julia.
- Out of the box not working on Julia-1.5.2(The default on Debian bullseye, the most recent Debian version) due to a version issue with SparseIR.
Report
This is a very valuable contribution to the scipost codebases journal and I recommend publication after a few minor improvements to the paper.
Requested changes
1.) The installation section should be elaborated to make it easier for users that have never used julia. What is a REPL? How do I get into package mode?
2.) The authors have invested quite some time into the numerical stability of their algorithm, yet the determination of the optimal number of Matsubara frequancies to include for the construction of the Pick matrix works on the assumption that utilizing the cholesky decomposition for determining positive definiteness "just works", although it is hard problem (S.M. RUMP, VERIFICATION OF POSITIVE DEFINITENESS, 2006)
3.) Numerous typos, and improvements to the utilized grammar would be helpful, some I point out here, but I recommend additional proof-reading.
- demension: sample code in figure 1, and in Table 6
- pg. 7 "we prepare recursive algorithm"
- Formatting issue with Hankel Matrix on pg. 9
- pg. 11 ".. is a Nevannlina function, Schur algorithm..."
- cut-off is written inconsistently throughout the paper.
- Table 1: momets
- Table 1(and others) "The number of mesh in real axis"
- pg. 13: Furtheremore
- pg. 16: "However, the existence of true undetermined function ..."
- pg. 18: Nevalinna
Author: Kosuke Nogaki on 2023-09-20 [id 3994]
(in reply to Report 1 on 2023-04-27)
We sincerely appreciate your comprehensive and constructive review. We are grateful for your positive comments and suggestions.
The paper and the software address an issue that has plagued the quantum Monte Carlo community for years and the authors have applied theory that exploits the analytic structure of the Matsubara Green'S function to improve the state of affairs. All the work and the obtained experiences have been made available to the public in a software package.
This is a very valuable contribution to the scipost codebases journal and I recommend publication after a few minor improvements to the paper.
In light of your suggestions, we have revised our manuscript thoroughly. We believe the revised version now meets the publication criteria of SciPost Physics Codebases.
The referee writes:
The installation section should be elaborated to make it easier for users that have never used julia. What is a REPL? How do I get into package mode?
Our response: We appreciate your suggestion. We have thus expanded Section 3.1 in our revised manuscript, providing a concise introduction to the Julia programming language and its package management system. Furthermore, we have incorporated more detailed instructions for the installation of necessary libraries.
The referee writes:
The authors have invested quite some time into the numerical stability of their algorithm, yet the determination of the optimal number of Matsubara frequencies to include for the construction of the Pick matrix works on the assumption that utilizing the cholesky decomposition for determining positive deficniteness "just works", although it is hard problem (S.M. RUMP, VERIFICATION OF POSITIVE DEFINITENESS, 2006)
Our response: We appreciate your insightful comment. We acknowledge that the numerical Cholesky decomposition does not necessarily provide a criterion for the positive definiteness of matrices. However, in our code, we use multiple precision arithmetic in the Cholesky decomposition, which significantly reduces the effect of rounding errors on our results. This point is further elaborated in Section 2.2.3 of the revised manuscript, and an in-depth discussion has been added to Section 4.
The referee writes:
Numerous typos, and improvements to the utilized grammar would be helpful, some I point out here, but I recommend additional proof-reading.
Our response: We appreciate your attention to detail in identifying typographical errors. The manuscript has been revised to incorporate your suggestions. Additional proofreading has been conducted to ensure the coherence and correctness of the manuscript. We are confident that our revised manuscript aligns with the publication standards of SciPost Physics Codebases.
Author: Kosuke Nogaki on 2023-09-20 [id 3995]
(in reply to Report 2 on 2023-05-04)We greatly appreciate your comprehensive and constructive review. We are grateful for your positive comments and suggestions.
Following your comments and suggestions, we have made extensive revisions to our manuscript. We believe that the revised version is now suitable for publication in SciPost Physics Codebases.
The referee writes:
Our response: In Section 3.3, we have added a discussion on the analytic continuation results obtained from both tractable and challenging Hubbard gap models. We have further explored the numerical instability observed in these cases and provided insights into its origin. We firmly believe that further investigations utilizing our code will significantly enhance the stability and capability of Nevanlinna analytic continuation.
The referee writes:
Our response: Thank you for your suggestion to make our software more user-friendly. We have added examples of δ-function, single-Gaussian, single-Lorentzian, Kondo-resonance, and Hubbard-gap models. Users can try out and reproduce these results using notebooks in the repository.
The referee writes:
Our response: Thank you for your suggestion. We have accordingly updated the notebooks in our repository The first cell of each notebook now provides information on the expected runtime, the chosen broadening parameter, and the list of required packages
The referee writes:
Our response: We have implemented a new interface, namely, a command line interface (CLI), which can be executed in a shell. This interface does not necessitate a deep understanding of data treatment in the Julia language from the users. Users only need to prepare input data as text files. We believe that this CLI is useful for users who are not experts in Julia.
The referee writes:
Our response: We have conducted a meticulous proofreading of our manuscript. We are confident that the revised version now meets the high standards of your publication.

---

## Round 1 · Referee Report · Anonymous · 2023-5-4

Strengths
- an open source software package for the analytic continuation method based on Nevanlinna functions
- uses strengths of the Julia language and associated libraries; in particular easy to use arbitrary precision and automatic differentiation
- an extension of the previous C++ implementation provided by the authors of the first paper on Nevanlinna analytic continuation
- numerical stability is much stronger than in previous versions
Weaknesses
- lacks novelty and critical assessment of the pros and cons
- too few examples. The Hubbard gap example is not worked out although it is provided in the repo.
- assumes in a too strong way that users are familiar with the Julia language
- installation and usage of the git repo required some changes on my platform
- too many spelling errors
Report
This is a very valuable contribution for Nevanlinna analytic continuation that will be of practical use to the community
Requested changes
1. work out the Hubbard gap example in the main text please. I could work through the notebook example in the git repo but that required quite some effort and does not tell me how the authors interpreted the data
2. More examples please.
3. Provide more information in the notebooks; eg on runtime, stability, judicious choice of parameters, installation
4. provide more interfaces (eg for other kernels) so that it is easy to use nevanlinna for non-experts and users that are not familiar with the julia language
5. the authors should make an effort to improve the many spelling errors

---

## Round 2 · Referee Report · Anonymous (Referee 1) · 2023-9-25

Strengths

  • The paper and the software address an issue that has plagued the quantum Monte Carlo community for years and the authors have applied theory that exploits the analytic structure of the Matsubara Green's function to improve the state of affairs.
  • All the work and the obtained experiences have been made available to the public in a software package.

Weaknesses

  • Opportunities for improvements have been taken.

Report

This is a very valuable contribution to the scipost codebases journal and I recommend publication after another round of proof-reading.

Requested changes

3 Usage
3.1 Installation
Firstly, users need to install Julia (v1.6 or newer) and make sure to add the location of the Julia executable (julia) "to your PATH environment variable."
-> "to their PATH environment variable."

---

## Round 2 · Referee Report · Anonymous (Referee 2) · 2023-10-11

Strengths

1 - an open source implementation of the Nevanlinna analytic continuation algorithm, which is relative new to the community and should be the method of choice for certain examples. This will be very valuable to the community
2 - various representative examples illustrate the algorithm and the package
3 - a clear description of the algorithm

Weaknesses

none

Report

I am satisfied with the changes in the repository and the manuscript.

Requested changes

none

---

## Round 2 · Author Response

Author comments upon resubmission

We sincerely thank you for sharing the highly positive feedback from the referees. We are thrilled to learn that our library is recognized as a valuable contribution to the community by both referees. We have individually addressed their comments and made extensive revisions to our manuscript based on their suggestions and feedback. We trust that our revised manuscript now adheres to the publication standards of SciPost Physics Codebases.

---

## Round 2 · List of Changes

List of changes
  1. We have implemented a command-line interface and explained its usage in Section 3.2.2 of our revised manuscript.
  2. The revised manuscript now includes a discussion on the rigorous criterion for positive definiteness in both Section 2.2.3 and Section 4.
  3. Figure 1 of the manuscript has been updated to include six examples of Nevanlinna analytic continuation.
  4. In Section 3.3, we have added a discussion on the challenging Hubbard gap model, which is presented by Figure 3.
  5. We have added references [56], [66], and [67] to the reference list.
  6. To enhance clarity, the sample code has been moved to Appendix B for easy reference.
  7. We have cleaned the notebooks in our repository.

---

## Editorial Decision

published